# OpenReview forum: "D-Judge: Disrupting Multi-Turn Jailbreaks using Semantics-Preserving Output Rewriting"
_ICML.cc/2026/Conference — ICML 2026 regular_

### Official Review · Reviewer_nWAB · 2026-03-10

**Soundness:** 3
**Presentation:** 3
**Significance:** 3
**Originality:** 2
**Overall Recommendation:** 4
**Confidence:** 3

**Summary:**

This paper addresses multi-turn jailbreak attacks on LLMs, where adversaries use an external "judge" LLM to iteratively score the victim model's responses and refine harmful queries. Rather than detecting or blocking harmful content at the final turn (as existing defenses do), the authors propose Decoy for the Judge (DJ), a semantics-preserving output rewriting system that intercepts the victim LLM's responses before they reach the attacker's judge, feeding the judge misleading harmfulness signals and corrupting the refinement loop. Two variants are trained: SFT-DJ (prioritizes semantic fidelity) and DPO-DJ (stronger defense, more aggressive rewriting).

**Compliance With Llm Reviewing Policy:**

Affirmed.

**Final Justification:**

I am convinced with the rebuttal and the follow-up answer of the authors. The ablation study of the dataset size in the experiments satisfies my concerns. I have increased my score from weak reject to weak accept.

**Key Questions For Authors:**

1. Why was Llama Guard chosen as the sole harmfulness oracle? Was any inter-annotator agreement or multi-model validation performed?
2. Are results in Table 3 artificially strong for Llama Guard specifically, due to training signal overlap?
3. Was any portion of PKU-SafeRLHF held out for independent dataset quality validation, separate from the model test split?
4. Was dataset size ablated? how much does performance drop with 50%, 25% of training data? Is there any ablation study for semantic threshold?
5. How is semantic equivalence validated beyond embedding similarity? Is there any human evaluation?

**Limitations:**

Yes

**Strengths And Weaknesses:**

The paper is technically sound, and the claims are generally well supported by the experimental results. The presentation is clear, well structured, and easy to follow. The authors address a well-motivated problem related to multi-turn jailbreak attacks on large language models (LLMs), in which adversaries leverage an external “judge” LLM to iteratively evaluate the victim model’s responses and refine harmful queries. The paper identifies an underexplored attack surface: the judge-driven feedback loop itself rather than the harmful output. This perspective is conceptually clean and represents a novel and interesting insight that is worthy of recognition.

The authors propose Decoy for the Judge (DJ), a defense mechanism that rewrites the victim LLM’s response before it reaches both the attacker and the judge model. Instead of directly blocking harmful responses, the system modifies the wording of responses while attempting to preserve their semantics, with the goal of misleading the judge model into misinterpreting them as more harmful or otherwise misleading. This idea of manipulating the feedback channel rather than the output channel is an original contribution.

The Decoy Dataset, which contains semantically similar responses with controlled harmfulness shifts (generated using INCREASE/DECREASE rewrites with LlamaGuard scoring and Sentence-BERT similarity filtering), is a concrete and potentially reusable resource. The paper evaluates the proposed defense across five multi-turn jailbreak attacks and reports promising results that support the paper’s claims. The authors also openly discuss several limitations and outline possible future research directions.

However, the Decoy Dataset is simultaneously the paper’s most novel contribution and its most fragile component. A compromised, biased, or overfitted dataset could silently undermine many of the downstream claims regarding transferability and defense effectiveness, and the paper currently provides limited evidence to fully rule out this concern.

First, the cosine similarity threshold used to enforce semantic preservation (τ = 0.8 based on Sentence-BERT embeddings) is a relatively coarse proxy for true semantic equivalence. Sentence-BERT similarity can fail to capture subtle meaning shifts, particularly in short or technical texts. The paper does not include human evaluation to validate whether the generated rewrites genuinely preserve semantics.

Second, the Decoy Dataset relies on LlamaGuard to label harmfulness shifts. At the same time, LlamaGuard is also used as an evaluation judge in Table 3. This creates a potential evaluation bias, as the defense may partially be evaluated using the same signal that was used during dataset construction and training.

Third, the dataset is generated entirely from the PKU-SafeRLHF dataset, which represents a specific distribution of harmful prompts and response styles. This raises concerns about distributional coverage. Real-world multi-turn jailbreak attacks may involve a broader range of topics and conversational structures that are not represented in PKU-SafeRLHF. As a result, the DJ method may experience performance degradation on out-of-distribution harmful content that was not present during dataset generation.

Finally, the robustness of the proposed defense in scenarios involving multiple or adaptive judges is not clearly established. If attackers employ multiple judge models simultaneously or use judges that are more robust to semantic rewriting, it is unclear whether DJ would remain effective.

---

> ### Author Rebuttal · Authors · 2026-03-30
>
> We thank the reviewer for the careful review and insightful comments.
>
> **Regarding semantic preservation:** We agree that a cosine-similarity threshold is only an approximate measure. To address this, we evaluated a stronger Semantic Gate using based on bidirectional entailment using a Natural Language Inference (NLI) model. Now the rewrite is accepted only if the original and rewritten texts mutually entail each other, rather than just having high embedding similarity. This provides a more meaningful check of semantic preservation than cosine similarity alone. The refined NLI-based gate improves benign-task performance while maintaining strong defense performance, increasing the average MT-Bench score from 6.02 to 8.34. We did not perform an ablation of semantic threshold since threshold selection is no longer required in this revised setup. Also, we agree that human evaluation would be valuable, but the discussion period is too short to conduct a careful study. If given the opportunity for a camera-ready version, we will add a study.
>
> **Regarding potential evaluation bias from Llama Guard:** We do not believe the Table 3 results are artificially strong for Llama Guard. During the dataset construction, Llama Guard is used only to measure relative harmfulness shifts between semantically aligned rewrites. In our evaluations, the attacker does not rely on Llama Guard for feedback: the judges are GPT-4o, GPT-5.2, and Gemini-3-Flash.
>
> Section 4.4 shows that the trained rewriters remain effective across multiple judge backbones and prompts, achieving even higher success ratios against other judges rather than Llama Guard in several cases, which argues against overfitting to Llama Guard-specific artifacts. We therefore view Llama Guard in Table 3 mainly as a diagnostic metric aligned with the training signal, while the cross-judge attack results provide evidence that the method generalizes beyond Llama Guard.
>
> **Regarding distributional coverage:** We agree that out-of-distribution generalization is an important concern. However, our evaluation already goes well beyond the distribution used to construct the dataset. We test DJ on five multi-turn, two single-turn jailbreak attacks, and benchmark datasets such as HarmBench and AdvBench, which cover a much broader range of topics and interaction styles than PKU-SafeRLHF alone. These attacks and benchmarks are distinct from PKU-SafeRLHF and are not used during dataset construction. The fact that DJ remains effective across these settings suggests that it generalizes beyond the specific distribution used to build the training data.
>
> **Regarding multiple or adaptive judges:** We agree that this is an important setting. Although we are not aware of established jailbreak attacks driven by a fully adaptive multi-judge setup, our results in Table 3 and 4 already suggest that DJ is not limited to weak or easily influenced judges. In addition, we evaluate the rewriter against stronger judge variants, including JailJudge, which uses a multi-agent framework for more comprehensive jailbreak evaluation, and StrongREJECT, which uses a stricter rubric-based evaluator designed to be less sensitive to superficial cues. The table below reports the fraction of rewritten responses that successfully change the judge score.
> |Judge|Success ratio (%)|
> |---|---:|
> |JailJudge-multi_agent|67.6|
> |JailJudge-FT|81.5|
> |StrongREJECT-rubric|80.4|
> |StrongREJECT-FT|86.8|
>
> **Regarding the choice of Llama Guard:** We chose Llama Guard because the Decoy Dataset requires a dense scalar signal that can capture fine-grained relative harmfulness shifts between semantically similar rewrites. We found that Llama Guard provides the best unsafe-probability signal for this purpose. We did not conduct a formal inter-annotator agreement study or multi-model ensemble labeling during dataset construction. However, our goal is not to fit a single judge’s absolute scores, but to learn from relative harmfulness orderings between semantically aligned rewrites. We will clarify this more clearly in the revision.
>
> **Regarding dataset size ablation:** We evaluated how performance changes with reduced training data. We evaluate the rewriter in isolation by comparing the base rewriter without training (0% data) against versions trained on 25%, 50%, and 100% of the training set. The success ratio of rewrites that flip the Llama Guard decision while preserving semantics increases from 48.9% to 57.7%, 58.8%, and 62.0%, respectively.
>
> **Regarding held-out validation within PKU-SafeRLHF:** We did not create an additional hold-out split specifically for dataset-quality validation. Our reason is that PKU-SafeRLHF is used here as a source dataset for constructing training data, not as the main benchmark for our claims. Instead, the main evaluation is carried out on independent downstream settings, including five multi-turn, two single-turn jailbreak attacks, and additional judge and victim models outside the PKU-SafeRLHF pipeline

---

> > ### Author Rebuttal · Reviewer_nWAB · 2026-04-03
> >
> > Thanks for the rebuttal. The rebuttal has addressed most of my concerns satisfactorily. However, I have some follow-up questions for the authors.
> >
> > The dataset size ablation still measures the wrong metric. Reporting LlamaGuard-flip success ratios as the ablation outcome for data scaling does not tell us whether fewer training samples would produce comparable ASR reduction. A correction of the dataset size ablation to report downstream ASR rather than LlamaGuard-flip rates would change my score. The authors need to commit to including in the final paper: the NLI gate results across all tables, the Circuit Breakers comparison with attack-specific discussion, the IFEval results, and an honest acknowledgment of the transferability limitation if accepted.

---

> > > ### Author Response · Authors · 2026-04-04
> > >
> > > We thank the reviewer for this important clarification. We report below the impact of training-data scale on downstream ASR under the same setup as Table 1, using the NLI-based Semantic Gate. The 0% setting corresponds to the base prompt rewriter without SFT or DPO.
> > > | % of data | Crescendo (↓) | CoA (↓) | FITD (↓) | ActorAttack (↓) | X-Teaming (↓) | Average (↓) |
> > > |---|---:|---:|---:|---:|---:|---:|
> > > | 0%   | 63.5 | 64.8 | 15.7 | 30.8 | 57.2 | 46.4 |
> > > | 25%  | 22.0 | 22.6 | 10.1 | 8.2  | 21.3 | 16.8 |
> > > | 50%  | 11.9 | 14.5 | 13.2 | 3.2  | 16.4 | 11.8 |
> > > | 100% | 8.8  | 8.2  | 9.4  | 1.9  | 14.5 | 8.6  |
> > >
> > > This ablation shows that fewer training samples already recover a substantial portion of the downstream defense benefit, while more data consistently yields stronger overall ASR reduction. In particular, using only 25% of the data reduces average ASR from 46.4 to 16.8, while 50% further reduces it to 11.8, and the full dataset reaches 8.6. The overall trend is clear and consistent: increasing training data improves downstream robustness.
> > >
> > > We are fully committed to incorporating these points in the camera-ready version if the paper is accepted. Specifically, we will:
> > >
> > >  (1) use the NLI gate as the default semantic-preservation check across all tables,
> > >
> > >  (2) include the Circuit Breakers comparison together with attack-specific discussion,
> > >
> > >  (3) include the IFEval results, and
> > >
> > >  (4) clearly acknowledge the current transferability limitation.
> > >
> > > We hope this resolves the reviewer’s remaining concern, and we would be very grateful if they would consider updating their score accordingly.

---

### Official Review · Reviewer_6C68 · 2026-03-12

**Soundness:** 3
**Presentation:** 3
**Significance:** 3
**Originality:** 3
**Overall Recommendation:** 4
**Confidence:** 3

**Summary:**

This paper proposes **Decoy for the Judge** (**DJ**), a semantics-preserving output rewriting framework that disrupts multi-turn jailbreaks by perturbing judge-driven feedback. **DJ** significantly reduces attack success rates while preserving benign performance.

**Compliance With Llm Reviewing Policy:**

Affirmed.

**Key Questions For Authors:**

Refer to weakness.

**Limitations:**

Yes.

**Strengths And Weaknesses:**

**Strengths:**

1. Mechanism: Targets the critical feedback loop of multi-turn jailbreaks. DJ rewrites responses to perturb judge scores while preserving semantics, directly disrupting the attack's optimization signal. This is critical for effectively mitigating multi-turn jailbreak attacks.
2. Performance: DJ substantially outperforms strong baselines, showing robust defense and generalization across diverse attack methods. Such performance and generalization stem from perturbing the critical step of multi-turn jailbreak attacks.
3. Presentation: Well-written, well-organized and clear, with no apparent errors or logical inconsistencies.

**Weaknesses:**

1. Transferabe Attack: Given the cross-model transferability of many attack methods, it remains unclear whether DJ remains effective when attackers optimize harmful samples on alternative models.

2. Quality Degradation: Experimental results indicate that rewriting degrades response quality. Could this lead to error accumulation in benign multi-turn dialogues?

3. Baselines: Should comparisons include a baseline evaluating the inherent safety of mainstream LLMs (e.g., Llama 3) against multi-turn jailbreaks?

---

> ### Author Rebuttal · Authors · 2026-03-30
>
> We thank the reviewer for the careful review and insightful comments.
>
> **Regarding transferable attacks:** We agree that this is an important and practical setting. DJ is mainly designed to disrupt the online refinement process used by multi-turn jailbreak attacks. In other words, its main target is not transferability itself, but the iterative optimization loop that produces increasingly effective harmful prompts. Consistent with this, when harmful samples are optimized against a DJ-protected victim using FITD, which has been reported to transfer across models, the resulting prompts achieve only 9.6% ASR even when they are reused against the same undefended victim model. This is much lower than the transferability numbers reported in the original FITD paper. These results suggest that prompts optimized under DJ are already strongly degraded, even in this easier reuse setting. A broader evaluation across different attacker-victim model pairs is beyond the scope of this paper, but we agree that this is an important direction for future work.
>
> **Regarding quality degradation and error accumulation:** We agree that this is an important question. DJ does not introduce an additional error-accumulation mechanism by design, because it rewrites each turn independently. The semantic gate also helps limit semantic drift at each step. As a result, any remaining degradation in long benign conversations is more likely to come from the underlying victim model’s handling of dialogue history, rather than from accumulation within DJ itself. In our additional benign multi-turn experiments with the refined semantic gate, we did not observe noticeable error accumulation. That said, we agree that preserving response quality over long benign interactions is an important direction for future work.
>
> **Regarding inherent-safety baselines:** We agree that this is an important comparison. The closed-source victim models in our evaluation, including GPT-4o, GPT-5 mini, and Gemini 3 Flash (Appendix B.2), already include built-in safety alignment. We will also add results on Llama 3.3, where DJ reduces the ASR of X-Teaming from 71.1% to 15.7%. This shows that DJ remains effective even when the victim model already has its own safety training.
>
> To provide a more direct comparison to an inherent-safety-style defense, we utilize the Llama-3 model with circuit breakers inserted using Representation Rerouting [1] as a standalone guardrail given our black-box setting. Under the same setup as Table 1, this defense achieves an average ASR of 12.0, compared with 8.6 for DPO-DJ with the refined gate.
>
> | Methods | Crescendo (↓) | CoA (↓) | FITD (↓) | ActorAttack (↓) | X-Teaming (↓) | Average (↓) |
> |---|---:|---:|---:|---:|---:|---:|
> | Circuit breaker | 11.3 | 24.1 | 1.2 | 3.1 | 20.1 | 12.0 |
> | DPO-DJ with refined gate | 8.8 | 8.2 | 9.4 | 1.9 | 14.5 | 8.6 |
>
> At the same time, DJ works entirely at the interaction boundary, without retraining, weight editing, or access to hidden activations, so it should be viewed as complementary to inherent-safety approaches rather than as a replacement.
>
>
> ---
> [1] A. Zou, L. Phan, J. Wang, D. Duenas, M. Lin, M. Andriushchenko, R. Wang, Z. Kolter, M. Fredrikson, and D. Hendrycks, “Improving Alignment and Robustness with Circuit Breakers,” arXiv preprint arXiv:2406.04313, 2024.

---

### Official Review · Reviewer_78wt · 2026-03-12

**Soundness:** 2
**Presentation:** 3
**Significance:** 2
**Originality:** 3
**Overall Recommendation:** 4
**Confidence:** 4

**Summary:**

In this paper, authors propose a novel defense mechanism named "Decoy for the Judge" (DJ) against multi-turn jailbreak attacks. The authors identify that these attacks rely on a feedback loop where an external judge LLM scores intermediate responses to help refine the attacker's queries. DJ disrupts this loop by deploying a rewriter LLM that alters the victim model's response before the attacker receives it. This rewrite preserves the semantic meaning of the response but subtly changes surface cues to artificially inflate the harmfulness score assigned by the attacker's judge. The authors construct a "Decoy Dataset" and train two variants of the defense: SFT-DJ (Supervised Fine-Tuning) and DPO-DJ (Direct Preference Optimization).

**Compliance With Llm Reviewing Policy:**

Affirmed.

**Final Justification:**

Based on the rebuttal and additional analysis provided, I have increased the score.

**Key Questions For Authors:**

1. DJ adds latency during inference time due to rewrites. What is a quantitative estimate of this overhead (e.g., average added latency in milliseconds or additional token generation costs) during a standard multi-turn interaction?

2. To mitigate the severe performance degradation on benign tasks seen with DPO-DJ, you suggest pairing DJ with a risk-estimating detector to adaptively choose the rewriting strength. Do you have any preliminary data or a specific architectural proposal for how this thresholding would operate in practice?

3. The Decoy Dataset is constructed by evaluating candidate rewrites using Llama Guard to compute the unsafe probability shifts (. Does relying on a single guard model (Llama Guard) inject specific biases into the rewriter?

**Limitations:**

yes

**Strengths And Weaknesses:**

Strengths
1. Paper presents an innovative idea of Decoy Judge that cleverly rewrites LLM’s responses, which uniquely intervenes directly in the attacker's feedback optimization loop.

2. The authors successfully demonstrate DJ's transferability across different judge configurations (GPT-4o, GPT-5.2, Gemini-3-Flash) and various scoring scales

Weaknesses

1. The most effective defense variant, DPO-DJ, comes at a high cost to the model's benign capabilities. On the MT-Bench evaluation, the average score drops from 8.99 (baseline) to 6.02 with DPO-DJ. While SFT-DJ preserves performance better (8.42), it offers less robust security (16.6% ASR).

2. Inserting a rewriter LLM into the API boundary (acknowledged by authors)  naturally increases both computational cost and latency. The paper does not quantify this overhead (e.g., token generation delay, added compute cost), which is a critical factor for real-world deployment.

3. Not sure if it is a fair comparison to decompose each attack into query generation and query update stages (Section 4.2).  Wonder how much the numbers deviated due to not using the best strategies reported in the papers.

---

> ### Author Rebuttal · Authors · 2026-03-30
>
> We thank the reviewer for the careful review and insightful comments.
>
> **Regarding the benign-utility tradeoff:** We agree that this is an important point. Motivated by this comment, we evaluated a refined Semantic Gate based on bidirectional entailment using a Natural Language Inference (NLI) model. This modification greatly reduces the benign-performance drop for the DPO-based variant.
> * Average MT-Bench score improves from 6.02 with the simple gate to 8.34 with the NLI-based gate.
> * Defense performance remains strong, with average ASR increasing only modestly from 6.5 to 8.6.
>
> |Methods|Coding|Extraction|Humanities|Math|Reasoning|Roleplay|STEM|Writing|Average|
> |---|---:|---:|---:|---:|---:|---:|---:|---:|---:|
> |Baseline |8.55|9.38|9.95|6.80|9.00|8.90|9.70|9.65|8.99|
> |DPO-DJ with simple gate| 8.10|7.45|4.85|6.35|6.00|4.35|6.60|4.48|6.02|
> |DPO-DJ with NLI gate|8.70|8.95|8.65|6.80|8.05|8.20|9.10|8.25|8.34|
>
> |Methods|Crescendo|CoA|FITD|ActorAttack|X-Teaming|Average|
> |---|---:|---:|---:|---:|---:|---:|
> |No Defense |79.9|89.9|32.7|54.1|78.0|67.0|
> |DPO-DJ with simple gate|5.0|4.4|7.6|0.6|15.1|6.5|
> |DPO-DJ with NLI gate|8.8|8.2|9.4|1.9|14.5|8.6|
>
> These results suggest that the utility loss is not inherent to DJ itself, but is mainly caused by limitations of the semantic filtering component. Using a better gate leads to a better security-utility tradeoff, while leaving the overall DJ framework unchanged.
>
> **Regarding computational cost and latency:** We agree that the rewriter introduces extra compute cost and latency (as pointed out in our limitation section). In our setup (vLLM, BF16), the rewriter runs at 5.4 ms/token. On MT-Bench, the average number of rewrites needed to pass the gate is 1.5, corresponding to an average additional per-turn latency of 97 ms. Overall, DJ does introduce additional inference overhead, but in our setup the overhead remains moderate because it uses a locally deployed 4B rewriter and a small gate model rather than a larger or commercial model. Whether this tradeoff is acceptable depends on the deployment setting and the desired level of protection.
>
> **Regarding the standardized attack decomposition:** Our goal in Section 4.2 is not to reproduce the best absolute ASR reported, but to place different attacks in a common judge-guided framework so that they can be compared more directly. This standardization does not make the attacks weaker. For example, when we run X-Teaming in its original configuration it achieves 71.1% ASR, which is lower than in our setting, and DJ still reduces this to 13.2%. We also find that Crescendo and CoA can reach higher ASR in our setup. Taken together, these results suggest that the decomposition is a controlled choice for fair comparison, rather than the reason for DJ’s gains.
>
> **Regarding adaptive thresholding with a risk-estimating detector:** This is a good question. We do not yet have preliminary experimental results for this adaptive setup. However, our updated results with the NLI-based Semantic Gate suggest that the benign-utility tradeoff can be improved with a better filtering component. One simple design would be to place a lightweight risk detector before DJ and use its score to choose the defense strength. For example, low-risk queries could use a weaker rewriting setting or even bypass rewriting, while higher-risk queries could trigger stronger rewriting, a stricter semantic threshold, or a larger retry budget.
>
> **Regarding potential bias from Llama Guard:** Llama Guard is used only during dataset construction to measure relative changes in unsafe probability between semantically similar rewrites, rather than as a deployment-time judge or as a fixed target for the defense. As a result, the rewriter is trained on pairs of semantically aligned rewrites with different harmfulness rankings, not on raw scores from a single guard model. We will clarify this more clearly in the revision.
>
> We also explored other judges during dataset construction, including Qwen Guard and GPT-based judges. We found that Llama Guard’s unsafe probability was best aligned with human intuition. Importantly, if the rewriter were mainly learning Llama Guard-specific quirks, it would not generalize across other judges. In Section 4.4 we show that the rewriters outperform strong commercial baselines, and DJ reduces ASR when using GPT-4o, GPT-5.2, and Gemini-3-Flash as judges.
>
> We also evaluate benign performance on IFEval: DPO-DJ with the NLI gate performs nearly identically to the baseline, while another output rewriter causes a large drop. If the rewritten outputs were biased, they would introduce systematic changes to the output structure, which is not observed. We report strict and loose accuracies at the prompt and instruction levels; higher is better.
> |Methods|Strict Prompt|Strict Instruction|Loose Prompt|Loose Instruction|
> |---|---:|---:|---:|---:|
> |Baseline |0.77|0.84|0.80|0.86|
> |Aligner|0.39|0.49|0.41| 0.50|
> |DPO-DJ with NLI gate|0.77|0.83|0.79|0.85|

---

> > ### Author Rebuttal · Reviewer_78wt · 2026-04-06
> >
> > I thank the authors for detailed response and clearly answering the questions. I have updated the scores accordingly.

---

> > > ### Author Response · Authors · 2026-04-07
> > >
> > > We sincerely thank the reviewer for the support of this work. These constructive feedbacks have significantly strengthened the quality of the paper.

---

### Official Review · Reviewer_YXTF · 2026-03-17

**Soundness:** 3
**Presentation:** 3
**Significance:** 3
**Originality:** 3
**Overall Recommendation:** 4
**Confidence:** 4

**Summary:**

This paper proposes a jailbreak defense method targeting attack algorithms that rely on an auxiliary judge model to iteratively refine inputs based on a victim model’s responses. The feedback loop is interrupted by introducing a rewriting mechanism that modifies the victim model’s responses, making them remain benign but appear more harmful to the judge. As a result, the attack algorithm is steered toward generating less harmful outputs, thereby reducing the overall effectiveness of the jailbreak. The approach involves training a rewrite model on a customized dataset, one only using SFT and the other followed with DPO. Experiments across both single-turn and multi-turn attack settings demonstrate the effectiveness of the proposed defense.

**Compliance With Llm Reviewing Policy:**

Affirmed.

**Key Questions For Authors:**

Does this imply that a straightforward way to circumvent the proposed defense is to modify the terminal condition of the attack? For example, by selecting the query that yields the lowest judge score? If so, this could potentially undermine the effectiveness of the defense.

**Limitations:**

yes

**Strengths And Weaknesses:**

**Strength:**
1. The paper is well written and easy to follow. The problem is clearly motivated, and most claims are supported by experimental results.

2. Limitations, particularly regarding benign performance, are appropriately discussed.

3. The experimental results show that the proposed rewriter mechanism significantly reduces attack success rate (ASR) compared to the baselines.

**Weakness:**
1. An important baseline appears to be missing to properly assess the contribution of SFT/DPO over a vanilla prompt rewriter. Including such a comparison would help isolate the gains from the training procedure.

2. The proposed defense may be easily circumvented. (See questions)

---

> ### Author Rebuttal · Authors · 2026-03-30
>
> We thank the reviewer for the careful review and insightful comments.
>
> **Regarding the baseline comparison:** We agree that this is an important baseline for isolating the contribution of SFT and DPO over a base prompt rewriter. Because the full DJ pipeline also includes the Semantic Gate, an end-to-end comparison alone does not cleanly separate gains from the rewriter and gains from filtering. Thus, we evaluate the rewriter in isolation.
> * Specifically, the 0% data setting corresponds to the base prompt rewriter without SFT and DPO.
> * In addition we show versions trained on 25%, 50%, and 100% of the training set.
> * The success ratio of rewrites that flip the Llama Guard decision while preserving semantics increases from 48.9% to 57.7%, 58.8%, and 62.0%, respectively.
>
>
>
> | % of data | 0% (base rewriter) | 25% | 50% | 100% |
> |---|---:|---:|---:|---:|
> | Success ratio | 48.9 | 57.7 | 58.8 | 62.0 |
>
> This indicates that the improvement comes from training the rewriter, rather than from the base rewriter alone.
>
> **Regarding the terminal condition:** Changing the terminal condition can partially weaken DJ, but does not eliminate its effect. We study this directly in Section 4.5. Our results show that when the attack continues optimizing instead of stopping early at the judge-score threshold, DJ becomes less effective and the ASR increases. However, the ASR remains substantially below the no-defense baseline. This suggests that DJ does not depend only on the stopping rule. Its main effect is to interfere with the attack’s iterative refinement process, making further optimization less effective. Overall, these results show that DJ remains useful even when the attack is modified in this way.
>
> **Regarding selecting the query with the lowest judge score:** We are not aware of established multi-turn attacks that select queries in this way. Existing judge-guided attacks typically use the judge score as a signal for progress toward a more successful attack, as described in Eq. (5). More importantly, DJ does not depend on pushing the score in only one direction.
>
> As discussed in Section 4.6, we also evaluate a *DECREASE* rewriter that lowers the perceived harmfulness score while preserving semantics, and this variant still reduces ASR under X-Teaming. This suggests that DJ works by making the attacker’s feedback signal less reliable, rather than simply by increasing scores. It is also possible that alternating between both rewrite directions would make score-based query selection even less reliable for the attacker. We agree that this is an interesting direction for future work.

---

> > ### Author Rebuttal · Reviewer_YXTF · 2026-04-06
> >
> > Thank you very much for the response. My questions and concerns are mostly addressed.

---

> > > ### Author Response · Authors · 2026-04-07
> > >
> > > We are pleased to know that we have fully resolved all questions and concerns. We sincerely thank the reviewer for the insightful feedbacks and for the appreciation of this work.

---

### Decision · Program_Chairs · 2026-04-30

**Decision:**

Accept (regular)

**Comment:**

This paper proposes a new defense against multi-turn LLM jailbreaking. The proposed method rewrites the output of the victim LLM so that the judge model cannot provide accurate feedback to attackers. The approach requires training a rewrite model on a customized dataset using either SFT or DPO.

Reviewers appreciate the clear motivation of the problem, the significant reduction in attack success rate (ASR), and the demonstrated transferability of DJ across different judge configurations.

During the rebuttal, the authors provided extensive explanations and new experimental results on the benign-utility tradeoff, comparisons to inherent-safety baselines, stronger judge variants, and the impact of training-data scale.